# Biosensors with Metal Ion–Phosphate Chelation Interaction for Molecular Recognition

**DOI:** 10.3390/molecules28114394

**Published:** 2023-05-28

**Authors:** Xiaohua Ma, Yuanqiang Hao, Xiaoxiao Dong, Ning Xia

**Affiliations:** 1Henan Key Laboratory of Biomolecular Recognition and Sensing, Shangqiu Normal University, Shangqiu 476000, China; maxiaohua@sqnu.edu.cn (X.M.); haoyuanqiang@aliyun.com (Y.H.); 2College of Chemistry and Chemical Engineering, Anyang Normal University, Anyang 455000, China; 13253013189@163.com

**Keywords:** molecular recognition, phosphate, biosensors, metal ion, Phos-tag, metal–organic frameworks

## Abstract

Biosensors show promising prospects in the assays of various targets due to their advantages of high sensitivity, good selectivity and rapid response. Molecular recognition is a key event of biosensors, which usually involves the interaction of antigen–antibody, aptamer–target, lectin–sugar, boronic acid–diol, metal chelation and DNA hybridization. Metal ions or complexes can specifically recognize phosphate groups in peptides or proteins, obviating the use of biorecognition elements. In this review, we summarized the design and applications of biosensors with metal ion–phosphate chelation interaction for molecular recognition. The sensing techniques include electrochemistry, fluorescence, colorimetry and so on.

## 1. Introduction

As an important part of chemical sensors and biosensors, recognition elements may significantly influence analytical performance in terms of sensitivity, selectivity, stability, and cost-effectiveness [1,2]. Classical recognition elements (e.g., enzymes, nucleic acids, antibodies and cells) have been used in a wide range of fields for a long time [3]. For example, antibody-based immunosensors have been applied for the quantification of different antigens changing from small molecules to tumor cells [4]. Cell biosensors have been developed to determine the general metabolic status of living organisms, such as bacteria, yeast, fungi and tissue culture cells [5,6]. Recent decades have witnessed the progress of artificial recognition elements in chemical sensors and biosensors, including aptamers (e.g., DNA, RNA or peptides), molecularly imprinted polymers, and affibodies [7,8,9,10]. Despite the successful applications of these recognition elements, there are still some intrinsic disadvantages, such as laborious and expensive processes, weak long-term stability, and relatively low batch-to-batch viability. Therefore, the exploration of new recognition elements is of great importance for the advancement of novel chemical sensors and biosensors.

Biomolecules always exhibit several typical but exclusive characteristics due to the physiochemical properties of their components, such as high isoelectric points, glycans on glycoproteins, phosphate groups on phosphorylated proteins/peptides and nucleotides, and exposed thiols on proteins [11,12]. These characteristics allow molecular recognition in bioassays based on physical or chemical interactions. For example, Zhou’s group reported the electrochemical detection of p53 proteins with maleimide/ferrocene (Fc)-capped gold nanoparticles to recognize the targets by maleimide-thiol interactions [13]. Various boronic acid-functionalized molecules and materials have been used to isolate and detect glycoproteins based on the formation of robust and reversible boronate ester bonds [14,15,16,17,18]. Phosphate groups on the surface of phosphorylated proteins/peptides and in the backbone of nucleic acids can be used as recognition sites. Different molecules and materials have been employed to identify phosphorylated proteins, such as carboxypeptidase Y, phosphorylated amino acid-specific antibodies, protein-binding domains, metal complexes and metallic oxides [19,20,21,22,23]. Among them, the specific interaction between metals or metal complexes and phosphates has attracted extensive attention for the isolation, immobilization and determination of phosphate-containing biomolecules [24,25].

Recently, many novel detection methods have been developed for the detection of phosphate group-containing biomolecules, including fluorescence, colorimetry, electrochemistry, photoelectrochemistry (PEC) and electrochemiluminescence (ECL) methods. At this point, it is of great importance to systematically summarize the advancement of biosensors based on metal ion–phosphate chelation interactions. To date, only a few reviews have introduced the development of such biosensors in certain chapters. For example, Zhou et al. summarized the achievement of bioanalysis of epigenetic modifiers, including DNA methylation and protein phosphorylation [26]. Advances in Phos-tag-based methodologies for the separation and detection of the phosphoproteome have been reviewed by Kinoshita’s group and Shirakawa’s group [27,28,29,30]. In this review, we aim to systematically summarize the development of biosensors with metal ion–phosphate chelation interaction for molecular recognition. This review is classified into three parts according to the type of recognition elements toward phosphate-containing targets: Phos-tag, metal ions and metal–organic frameworks (MOFs). We believe that this article will stimulate broader interest in developing more powerful and innovative biosensors for the benefit of human health analysis.

## 2. Phos-Tag

The dinuclear Zn(II) complex of 1,3-bis[bis(pyridin-2-ylmethyl)-amino]propan-2-olato, named Phos-tag, has been considered to be a prevalent phosphate-binding tag. The complex, with a vacancy on the two Zn(II) ions, can specifically chelate with phosphate monoester dianion in the phosphorylated peptide or protein. The formed 1:1 complex shows a dissociation constant of approximately 10^−8^ M at a neutral pH value [31]. The affinity for phosphate monoester dianion is more than 10,000 times stronger than that for other anions. The value is close to that of the interaction between an anti-phosphorylation antibody and phosphorylated amino acid residues. Because of its small size, Phos-tag shows negligible effect on the structure around the phosphorylation site. Nowadays, Phos-tag and its derivatives are commercially available for the capture, separation and labeling of biomolecules with phosphate groups due to their intrinsic advantages of facile synthesis, excellent specificity against phosphate, high stability and low cost [32,33,34,35,36,37,38]. Moreover, Phos-tag modified with other functional units (e.g., biotin, electroactive molecules and fluorescent dyes) can act as the bridge to link the signal label with the phosphorylated target site in bioassays [39,40].

### 2.1. Electrochemical Methods

Electrochemical methods have been proven to be simpler and more cost-effective than traditional techniques in various fields, including disease diagnosis, medical research, and food safety [41,42]. Phos-tag-biotin can be used to specifically label the target by binding to the phosphate group, followed by conjugation with avidin-modified signal labels, including electroactive molecules, enzymes and nanomaterials (Table 1). Heterogeneous assays exhibit high sensitivity, but the steric hindrance effect in such methods may limit the interaction between enzyme and substrate immobilized on the solid surface. For this consideration, we developed an electrochemical biosensor for the detection of protein kinase A (PKA) by integrating the advantages of homogeneous reaction and heterogeneous assay [43]. In this work, the phosphorylation reaction occurred in a homogeneous solution and the detection assay was carried out at a solid electrode surface. As illustrated in Figure 1, a biotinylated peptide in solution was effectively phosphorylated by PKA and the product was captured by the neutravidin (NA)-modified electrode through the avidin–biotin interaction. Then, Fc-capped Phos-tag-modified gold nanoparticles (AuNPs) were used to recognize the phosphate groups on the electrode, generating a strong electrochemical signal. By taking advantage of homogeneous reaction and heterogeneous detection, this method achieved a low detection limit (5 mU/mL) for PKA detection.

Silver nanoparticles (AgNPs) with excellent electrochemical reactivity have been widely used in electrochemical biosensors. Yin et al. developed an electrochemical immunosensor for N6-methyladenosine-5′-triphosphate (m^6^ATP) detection using Phos-tag-biotin and Ag-SiO_2_ as the signal amplification label [44]. As shown in Figure 2, after the capture of m^6^ATP by anti-m^6^ATP antibody, Phos-tag-biotin was added to recognize m^6^ATP. In the presence of streptavidin (SA), biotin-modified Ag-SiO_2_ was immobilized on the electrode and produced an electrochemical signal. The proposed immunosensor achieved a wide linear range from 0.2 to 500 nM and a detection limit of 0.078 nM.

Natural enzymes can catalyze various reactions and generate an amplified signal for electrochemical detection. Generally, when Phos-tag–biotin was bound with the phosphate group on a target, avidin-labeled enzyme specifically bound to biotin by the avidin–biotin coupling chemistry, thus catalyzing the corresponding reaction in the presence of substrates. For example, alkaline phosphatase (ALP) has been used to develop electrochemical biosensors for the detection of polynucleotide kinase (PNK) and 5-hydroxymethylcytosine, respectively [45,46]. *β*-Galactosidase can enzymatically hydrolyze *p*-aminophenyl galactopyranoside (PAPG) into the electroactive molecule *p*-aminophenol (PAP) in a neutral environment. Zhou et al. reported an electrochemical biosensor for PKA activity detection based on Phos-tag-biotin and *β*-galactosidase [47]. As shown in Figure 3, Phos-tag-biotin captures SA-modified SiO_2_ on the electrode and subsequently recruits more biotin-*β*-galactosidase-modified AuNPs via the specific biotin-SA interaction. The electrochemical signal of *β*-galactosidase-catalyzed product (PAP) indirectly reflects the activity of PKA. Additionally, Yin et al. developed a horseradish peroxidase (HRP)-based electrochemical biosensor for the detection of PKA activity (Figure 4) [48]. In this work, HRP catalyzed the oxidation of hydroquinone by H_2_O_2_, and the oxidative product of benzoquinone was electrochemically reduced to generate a reduction current. Benefitting from the enzymatic signal amplification, this method showed a linear detection range from 0.5 to 25 unit/mL with a detection limit of 0.15 unit/mL.

The PEC process is based on photo-to-electric conversion resulting from electron excitation and subsequent charge transfer of photoactive materials under illumination. The combination of the PEC process and electrochemical bioanalysis has boosted the development of numerous PEC methods due to their advantages of low background, low cost, simple instrumentation and high sensitivity [61,62,63]. Photoactive material is the essential part of PEC biosensors, and the interaction between the formed material and the photoactive material-modified electrode may result in a change in the PEC signal [49]. Wang et al. developed a dual-signal amplified Phos-tag-biotin-based PEC biosensor for the detection of m^6^ATP [64]. The target of m^6^ATP was captured by the anti-m^6^ATP antibody modified on the MoS_2_-Au/BiVO_4_@TiO_2_/ITO electrode and further labeled with Phos-tag-biotin. Then, the avidin-SiO_2_-hDNA@Ag bioconjugates were tethered onto the electrode. After the addition of HRP-modified complementary DNA, Ag^+^ ions were released and reduced by the superoxide ions generated by the HRP-catalyzed decomposition of H_2_O_2_. The deposited Ag on AuNPs led to an increased photocurrent. Finally, this dual-signal amplified biosensor showed a low detection limit of 1.665 pM for m^6^ATP.

Phos-tag-based biorecognition events can introduce functional materials into the biosensing system and result in the generation, enhancement or impairment of the PEC signal [50,51]. The biological recognition system itself can change the photocurrent. For instance, Zhou et al. demonstrated that SA immobilized on the electrode by Phos-tag-biotin blocks the diffusion of ascorbic acid (AA) to the Bi_2_S_3_ surface, resulting in an obvious decrease in the photocurrent [52]. However, the sensitivity of these methods is always relatively low.

To improve the sensitivity, Wang et al. reported a PEC immunosensor for the determination of m^6^ATP [53]. As displayed in Figure 5A, a large quantity of Ru(bpy)_3_^2+^ complexes were doped into avidin-modified silica nanoparticles (avidin-SiO_2_@Ru). When the nanocomposites were captured on the electrode by Phos-tag-biotin, Ru(bpy)_3_^2+^ complexes greatly enhanced the photocurrent in the presence of AA. The PEC immunosensor exhibited a linear range of 0.01–10 nM with a detection limit of 3.23 pM for m^6^ATP. Additionally, enzymes can catalyze the in situ production of electron donors to generate a photocurrent [65]. For example, Ai et al. reported a PEC biosensor for m6A detection based on the ALP-catalyzed production of AA [54]. Yin et al. developed a PEC biosensor for the detection of histone acetyltransferase and inhibitor screening using β-galactosidase to catalyze the generation of the PEC electron donor 4-aminophenol [55]. Li et al. reported the ALP-based PEC detection of miRNA-319a in rice leaf based on CuO-CuWO_4_ [56]. As presented in Figure 5B, the target of miRNA-319a was determined with the signal amplification of rolling-circle amplification (RCA) in the presence of a miRNA-319a-specific linear padlock DNA probe and dNTP. The generated DNA was further digested with Nb.BsmI nicking enzyme to generate a large number of short ssDNA strands that hybridized with the probe DNA on the electrode surface. After labeling of the 5′-terminal phosphate group with Phos-tag-biotin, avidin-ALP was captured by the electrode and catalyzed the hydrolysis of *L*-ascorbic acid 2-phosphate (AAP) into the electron donor AA for PEC detection.

ECL is one type of luminescence that occurs at/near the electrode surface due to electrochemical and chemiluminescent reactions. Because of the fast analysis speed, inexpensive equipment, and background-free light interference, ECL biosensors have shown great potential in advanced biological and chemical sensing, converting biological recognition events into ECL signals [66,67]. The Ru(bpy)_3_^2+^ complex and its derivatives have become popular luminescent reagents in ECL because of their high stability and strong ECL activity. Sui et al. developed a Ru(bpy)_3_^2+^ complex-based ECL biosensor for 5-hydroxymethylcytosine (5 hmC) detection [57]. As displayed in Figure 6, Fe_3_O_4_ nanospheres were functionalized with polydopamine and 4-mercaptophenylboronic acid (MPBA) (a). In the presence of DNA methyltransferase (M.HhaI), 5 hmC was linked to the Fe_3_O_4_ nanospheres based on the M.HhaI-catalyzed covalent bonding reaction of –CH_2_OH and thiol groups. Then, Phos-tag-biotin was added to label the phosphate group of 5 hmC for covalent modification of the Ru(bpy)_3_^2+^ complex, producing an ECL signal (b). However, the 1:1 ratio between the Ru(bpy)_3_^2+^ complex and Phos-tag-biotin dramatically limited the detection sensitivity. To overcome this problem, Jiang et al. used multi-branched polyamidoamine dendrimers to carry avidin and Ru(II) molecules for the ECL determination of 5-hydroxymethylcytosine with the aid of Phos-tag-biotin as the bridge [58].

### 2.2. Fluorescence Methods

As one type of homogeneous technique, fluorescence analysis has attracted extensive attention due to its particular advantages of easy readout, high-throughput capability and low sample volume. Fluorescence resonance energy transfer (FRET) is a distance-dependent interaction between two dye molecules in which excitation energy is transferred from a donor molecule to an acceptor/quencher molecule without emission of a photon. Koike’s group and Hong’s group have reported several Phos-tag-based FRET assays for the detection of the activities of protein kinase and ALP, respectively [68,69,70,71]. For example, Rhee et al. prepared two fluorescence quencher-modified Phos-tags for real-time fluorescence monitoring of Abelson tyrosine kinase activity with the peptide substrate conjugated with a matching fluorophore [72]. However, organic fluorescent dyes may suffer from poor photostability and low quantum efficiency. Because of the remarkable advantages of narrow size-tunable emission spectrum, excellent photostability and high quantum yield, quantum dots (QDs) have been used as fluorescent labels in bioassays and bioimaging. Takanobu et al. reported a FRET-based protein kinase assay using Phos-tag-modified QDs and dye-labeled peptides [73]. After phosphorylation and recognition, Cy5 in the substrate peptide quenched the fluorescence of QDs via a FRET process.

Magnetic beads (MBs) can act as biomolecular immobilization carriers to capture and separate target molecules from complex samples with the aid of a magnet. Therefore, MBs have been widely integrated with various biosensors to lower the background signal. Jiang et al. constructed a Phos-tag-based fluorescent magnetobiosensor for the simultaneous detection of multiple protein kinases [59]. In this work, a biotinylated Phos-tag was used to recognize the phosphorylated site in different dye-labeled substrate peptides. SA-coated MBs were added to capture phosphorylated peptides via the interaction between biotin and SA. After magnetic separation, the phosphorylated peptides were released from the surface of the MBs and then detected by steady-state fluorescence measurements for the determination of PKA and Akt1, respectively. To further enhance detection sensitivity, Jiang et al. reported on a Phos-tag-based fluorescent magnetobiosensor for sensitive detection of protein kinase activity based on peptide–DNA conjugate-mediated RNase HII-driven cycling signal amplification [60]. After the separation and release of phosphorylated peptide–DNA conjugates, the DNA domain in the conjugate hybridized with the signal probe modified with a fluorophore (Cy5) and a quencher (BHQ2), activating the RNase HII-catalyzed cleavage. Cy5 molecules were separated from the quencher, and the fluorescence was recovered. Then, the released peptide–DNA conjugates triggered other RNase HII-catalyzed cleavage, eventually leading to an amplified fluorescence signal.

## 3. Metal Ions

It has been reported that some transition and lanthanide metal ions such as Fe^3+^, Ga^3+^ and Dy^3+^ can selectively bind to phosphates on biomolecules when they are chelated with specific coordinating ligands [74,75]. Typically, the Zr^4+^ ion can form a 1:1 or 1:2 complex (–PO_3_^2−^–Zr^4+^− or –PO_3_^2−^–Zr^4+^–PO_3_^2−^) with phosphate-containing molecules. Such interactions have been widely used as bridges for molecule immobilization and biosensing (Table 2) [76,77,78,79].

### 3.1. Electrochemical Biosensors

Besides being used as recognition elements, metal ions with catalytic or electroactive properties can be directly used as the signal probe. For example, Wang et al. reported the label-free monitoring of PKA activity using Fe^3+^ ions as electrocatalysts [80]. In this study, Fe^3+^ ions electrochemically catalyzed the reduction of H_2_O_2_ when they bound to phosphorylated sites of peptide substrates, generating a strong electrocatalytic current response. In addition, Wieckowska et al. developed an electrochemical biosensor for the detection of casein kinase activity, in which Ag^+^ ions linked to phosphate residues produced an electrochemical reduction signal [81].

In situ assembly or polymerization of signal probes has been proven to be an effective approach to generating a strong response with small fluctuations in target concentrations [82]. Several groups have reported innovative electrochemical bioassays based on Zr^4+^-mediated chelation and electrochemically mediated atom transfer radical polymerization (eATRP) [83,84,85,86,87,88,89]. For instance, Hu et al. reported the electrochemical detection of DNA by using surface-initiated eATRP (SI-eATRP) [90]. As shown in Figure 7A, peptide nucleic acid (PNA) was immobilized on the electrode to capture target DNA (tDNA). After the hybridization, Zr^4+^ ions were added to link the ATRP initiators with the PNA/DNA hetero-duplexes via the phosphate-Zr^4+^-carboxylate chemistry. Under the SI-eATRP reaction, long polymeric chains with numerous ferrocenylmethyl groups were formed in situ on the electrode, producing an amplified electrochemical signal. To further improve the sensitivity, graphene oxide was used to load the initiators, eventually generating numerous polymeric chains labeled with electroactive ferrocene groups and producing an amplified signal [91]. Nevertheless, the introduction of transition metals in most ATRP reactions may result in environmental pollution. Recently, Hu et al. reported a metal-free rose bengal-mediated photoinduced ATRP (PhotoATRP) for the detection of target DNA [92]. In this study, the photoATRP reaction was triggered by the photocatalyst rose bengal and the electron donor *β*-nicotinamide adenine dinucleotide under excitation with blue light.

Zr^4+^ ions can also link DNA to the phosphorylated peptide based on the sandwich-like complex (–PO_3_^2−^–Zr^4+^–PO_3_^2−^). Thus, versatile DNA amplification techniques have been integrated into bioassays for secondary signal amplification [93]. For example, Cheng et al. reported the sensitive electrochemical detection of PKA activity based on a Zr^4+^-mediated hybrid chain reaction (HCR) [94]. As presented in Figure 7B, a phosphorylated peptide captured a DNA primer with a 5′-phosphoryl end in the presence of Zr^4+^ ions. Then, the HCR reaction was triggered in the presence of molecular beacons. After the addition of hemin, the generated HRP-mimicking DNAzymes catalyzed the redox reaction between hydroquinone and H_2_O_2_, producing an amplified electrochemical signal.

**Figure 7 molecules-28-04394-f007:**
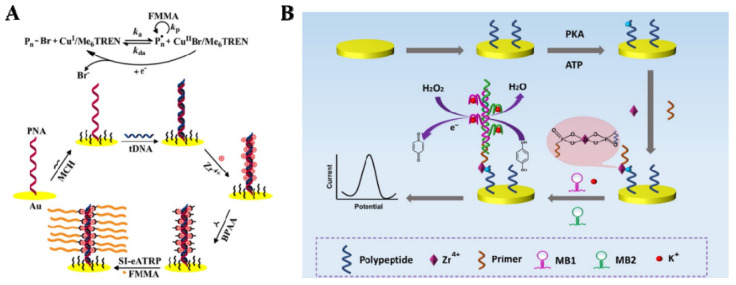
(**A**) Schematic illustration of Zr^4+^-mediated chelation and SI-eATRP-based electrochemical detection of DNA [90]. Copyright 2016 American Chemical Society. (**B**) Schematic illustration of the proposed strategy for PKA activity detection based on Zr^4+^-mediated HCR [94]. Copyright 2021 Elsevier.

Phosphate group-labeled DNA-modified nanomaterials can be anchored to the phosphorylated peptide or DNA on the electrode using metal ions as the linker. For example, Wang et al. reported the electrochemical detection of T4PNK activity based on DNA and aminoferrocene-modified single-wall carbon nanotube and Ti^4+^ ions [95]. Xu et al. used DNA-modified AuNPs to specifically label phosphorylated peptides with the aid of Zr^4+^ ions for the detection of PKA activity [96]. The self-assembled AuNP aggregates showed strong electroactivity and conductivity, enhancing the sensitivity of bioassays. Wang et al. developed an electrochemical biosensor for PKA activity analysis using DNA-assembled AuNP networks [97]. As illustrated in Figure 8, after the PKA-catalyzed phosphorylation, DNA-AuNPs were adsorbed on the phosphopeptide-modified electrode in the presence of Zr^4+^ ions. The DNA-AuNPs recruited more complementary DNA-modified AuNPs. Then, the formed AuNP networks absorbed more [Ru(bpy)_3_]^2+^ complexes, resulting in the significant enhancement of the current signal.

**Table 2 molecules-28-04394-t002:** Comparison of the detection performances of biosensors based on metal ion–phosphate chelation interaction.

Method	Substrate	Signal Report	Metal Ions	Target	Linear Range	Detection Limit	Ref.
EC	PNA	SI-eATRP of FMMA	Zr^4+^	DNA	1 × 10^−5^–10 pM	3.2 aM	[83]
EC	PNA	SI-eATRP of C_3_H_4_O/silver	Zr^4+^	DNA	1 × 10^−7^–1 nM	0.89 aM	[84]
EC	Peptide-AuE	eATRP of FMMA	Zr^4+^	PKA	0–1.4 × 10^2^ U/mL	1.63 U/mL	[85]
EC	Initiator	ARGET ATRP of FMMA	Zr^4+^	ALP	20–2 × 10^2^ U/mL	1.64 U/mL	[86]
EC	Peptide	RAFT of FMMA	Zr^4+^	DNA	1 × 10^−5^–1 nM	10.73 aM	[87]
EC	Peptide	RAFT of FMMA	Zr^4+^	PKA	0–0.14 U/mL	1.05 U/mL	[88]
EC	Peptide	TEMPO	Zr^4+^	DNA	1 × 10^−2^–1 × 10^2^ nM	2.57 pM	[89]
EC	PNA	SI-eATRP of FMMA	Zr^4+^	DNA	1 × 10^−5^– 0.1 nM	72 aM	[90]
EC	PNA	GO-eATRP of FMMA	Zr^4+^	DNA	1 × 10^−3^–1 × 10^2^ fM	0.213 aM	[91]
EC	PNA	Rose bengal-mediated photoATRP of FMMA	Zr^4+^	DNA	1–1 × 10^5^ fM	0.115 fM	[92]
EC	Peptide	pDNA + RCA	Zr^4+^	PKA	5–5 × 10^2^ U/mL	0.5 U/mL	[93]
EC	Peptide	pDNA + TMSDR/HCR	Zr^4+^	PKA	5 × 10^−2^–1 × 10^2^ U/mL	20 mU/mL	[94]
EC	DNA	psDNA + Fc-SWNTs	Zr^4+^	PNK	1 × 10^−2^–10 U/mL	10 mU/mL	[95]
EC	Peptide	pDNA-AuNPs	Zr^4+^	PKA	0–2.5 × 10^2^ U/mL	0.15 U/mL	[96]
EC	Peptide	DNA-AuNPs	Zr^4+^	PKA	0.1–40 U/mL	30 mU/mL	[97]
ECL	Kemptide-MBs	pDNA/TBR-cysteamine-AuNP	Zr^4+^	PKA	1 × 10^−2^–50 U/mL	5 mU/mL	[98]
ECL	Kemptide	pDNA/XOD-AuNPs	Zr^4+^	PKA	0.1–10 U/mL	90 mU/mL	[99]
ECL	Peptide-GQDs	pDNA/G-quadruplex–hemin DNAzyme- AuNPs	Zr^4+^	PKA	5 × 10^−2^–5 U/mL	40 mU/mL	[100]
ECL	Peptide	Ru(II)-SiO_2_ NPs	Zr^4+^	PKA	1 × 10^−2^–1 U/mL	5 mU/mL	[101]
PEC	Peptide-AuNPs	P-*g*-C_3_N_4_	Zr^4+^	PKA	5 × 10^−2^–50 U/mL	77 mU/mL	[102]
PEC	Kemptide TiO_2_	pDNA-AuNPs	Zr^4+^	PKA	8 × 10^−3^–1 U/mL	5 mU/mL	[103]
FL	–	phosphorylated and pyrene-labeled DNA	Zr^4+^	Zr^4+^	0.5–1 × 10^2^μM	200 nM	[104]
FL	–	FITC-peptide/polymer	Zr^4+^	PKA	0.5–1 × 10^3^ U/mL	0.2 U/mL	[105]
FL	SiO_2_	FITC-peptide	Zr^4+^	PKA	1 × 10^−2^–50 U/mL	6 mU/mL	[106]
FL	NTA-MNPs	FITC-peptide	Zr^4+^	PKA	Not reported	0.8 U/mL	[107]
FL	PNA-MBs	ATRP of FITC-o-acrylate	Zr^4+^	DNA	0.1 fM–0.1 nM	35.5 aM	[108]
FL	–	DNA-QDs/Cy5-peptide	Zr^4+^	PKA	3 × 10^−2^–100 U/mL	0.882 mU/mL	[109]
FL	–	Peptide−GQD	Zr^4+^	CK2	0.1–1 U/mL	30 mU/mL	[110]
FL	–	Peptide-AuNCs	Zr^4+^	CK2	8 × 10^−2^–2 U/mL	27 mU/mL	[111]
FL	Zr^4+^-MBs	peptide/avidin-UCNPs	Zr^4+^	PKA	5 × 10^−2^–0.2 U/mL	20 mU/mL	[112]
FL	DNA-MBs	TRITC-peptide	Dy^3+^	PKA	0.5–4 U/mL	0.12 U/mL	[113]
FL	–	TRITC-peptide/UCNPs	–	PKA	0.1–10 U/mL	50 mU/mL	[114]
Color	–	ATP-AuNPs	–	Zr^4+^	0.5–1 × 10^2^ μM	95 nM	[115]
Color	–	Citrate-AuNPs	Zr^4+^	ATP	0.1–15 μM	28 nM	[76]
Color	–	AuNPs	Zr^4+^	1B	5–1.8 × 10^2^ mU/mL	1.7 mU/mL	[116]
PGM	Zr^4+^-MBs	Invertase	Zr^4+^	PKA	0.1–10 U/mL	0.1 U/mL	[117]

Abbreviation: TEMPO, 2,2,6,6-tetramethylpiperidine 1-Oxyl; pDNA, phosphorylated DNA; RCA, rolling circle amplification; PKA, protein kinase A; TMSDR, toehold-mediated strand displacement reaction; HCR, hybridization chain reaction; PNA, peptide nucleic acid; SI-eATRP, electrochemically mediated surface-initiated atom transfer radical polymerization; FMMA, ferrocenylmethyl methacrylate; ARGET ATRP, activators regenerated by electron transfer atom transfer radical polymerization; AuNPs, gold nanoparticles; ALP, alkaline phosphatase; RAFT, reversible addition-fragmentation chain transfer; GO, graphene oxide; SWNTs, single-wall carbon nanotubes; PNK, T4 polynucleotide kinase; SiO_2_ NPs, silica nanoparticles; GQDs, graphene quantum dots; XOD, Xanthine oxidase; ITO, indium-tin oxide electrode; p-*g*-C_3_N_4_, phosphorylated graphite-like carbon nitride; MBs, magnetic beads; TBR, tris-(2,2′-bipyridyl) ruthenium; photoATRP, photoinduced atom transfer radical polymerization; UCNPs, upconversion nanophosphors; NTA, nitrilotriacetic acid; MNPs, magnetic nanoparticles; AuNCs, gold nanoclusters; CK2, casein kinase II; FITC, fluorescein; TRITC, tetramethylrhodamine.

Nanomaterials can act as carriers to load DNA and plenty of ECL indicators for recognition and signal output. For instance, Zhao et al. and Wang et al. reported ECL detection of PKA activity using 5′-phosphate group end DNA-conjugated AuNPs [98,99]. Recently, many nanomaterials have been successfully used as ECL indicators to replace traditional dyes. Liu et al. developed an ECL bioassay for PKA detection based on double-quenching of GQDs by G-quadruplex-hemin DNAzyme and AuNPs (Figure 9) [100]. In this work, GQDs were immobilized on the electrode and produced a strong ECL emission with the coreactant H_2_O_2_. After the phosphorylation and Zr^4+^-mediated assembly of DNAzyme-modified AuNPs, DNAzyme competitively decomposed the coreactant H_2_O_2_ and decreased the ECL intensity. Meanwhile, AuNPs quenched the ECL emission of GQDs through ECL resonance energy transfer. To avoid the complicated modification of DNA, Ru@SiO_2_ nanocomposites could be chemically phosphorylated and bound to the peptide by a Zr^4+^-based coordination interaction for the PKA activity assay [101].

PEC-active species, including molecules and nanomaterials, can be employed as signal labels or indicators to generate or enhance the electrical signal. For instance, graphite-like carbon nitride (P-g-C_3_N_4_) can be chemically phosphorylated and used to label phosphopeptides for PEC detection [102]. Yan et al. reported a visible-light PEC biosensor for the sensitive detection of PKA activity using phosphate-labeled DNA-modified AuNPs [103]. As shown in Figure 10, DNA-AuNPs were immobilized on the phosphorylated peptide-modified electrode with the assistance of Zr^4+^ ions. Many [Ru(bpy)_3_]^2+^ complexes adsorbed by DNA harvested visible light, and the excited electrons were injected into the conduction band of TiO_2_ to generate a photocurrent. Meanwhile, the localized surface plasmon resonance (LSPR) of AuNPs enhanced the photocurrent efficiency. This biosensor had excellent performance for inhibitor screening and PKA detection in cell lysates.

### 3.2. Fluorescence Biosensors

It has been documented that Zr^4+^ ions can link to two phosphate-containing molecules based on coordinate interaction, forming a sandwich-structured complex (–PO_3_^2−^–Zr^4+^–PO_3_^2−^–). For this view, Meng et al. reported the fluorescence assay of Zr^4+^ ions through the specific and robust interaction between phosphate groups and Zr^4+^ [104]. As illustrated in Figure 11A, the phosphorylated and pyrene-labeled DNA strands were used as the recognition and reporting units. The Zr^4+^ ion induces the formation of a hairpin structure and the two labeled pyrene molecules in close proximity, producing a strong excimer fluorescent emission. Then, γ-cyclodextrin (γ-CD) was introduced to modulate the space proximity via the cyclodextrin/pyrene inclusion interaction, resulting in an amplified fluorescence signal. This method showed a fluorescence “turn-on” response toward Zr^4+^ ions and had an increased sensitivity with a detection limit of 200 nM. Phosphopeptides modified with fluorescent dyes can be captured by metal ion-functionalized nanomaterials for subsequent fluorescence measurements [105,118]. Ren et al. reported a mix-and-read cytometric bead assay for the detection of PKA activity using Zr^4+^-functionalized mesoporous SiO_2_ microspheres (ZrMMs) (Figure 11B) [106]. In the presence of PKA and adenosine triphosphate (ATP), a fluorescein (FITC)-labeled peptide was phosphorylated and adsorbed on the surface of the ZrMMs through the binding between Zr^4+^ ions and phosphate groups. The fluorescence signal of FITC accumulated on ZrMMs was recorded using a flow cytometer, indirectly reflecting the PKA activity. Zr^4+^-immobilized nitrilotriacetic acid (NTA)-coated magnetic nanoparticles (Zr-NTA-MNPs) can also be used to specifically isolate and enrich dye-labeled phosphopeptides with a magnetic field. Tan et al. reported fluorescent detection of PKA based on Zr-NTA-MNPs [107]. As shown in Figure 11C, after the phosphorylation catalyzed by PKA, the FITC-labeled phosphorylated peptide was adsorbed on Zr-NTA-MNPs via the chelation of Zr^4+^ and phosphate groups. With the aid of an extra magnet, Zr-NTA-MNPs were easily separated from the homogeneous solution, resulting in decreased fluorescence intensity. In situ assembly or polymerization of dyes can greatly amplify the fluorescence signal for sensitive detection of low-abundance biomolecules. Zhang et al. reported a novel fluorescence biosensor for lung cancer DNA detection based on phosphate-Zr^4+^-sulfonate chemistry and atom transfer radical polymerization (ATRP) [108]. As presented in Figure 11D, peptide nucleic acid (PNA) modified on MBs hybridizes with target DNA (tDNA) and adsorbs Zr^4+^ and sulfonate to form the phosphate-Zr^4+^-sulfonate complex. In the presence of a catalyst and fluorescein-oacrylate (FA) monomer, the ATRP reaction was initiated to form polymer chains with multiple FAs.

Nowadays, plenty of fluorescent nanomaterials have been successfully prepared to replace traditional organic dyes for protein kinase detection, including gold nanoclusters (AuNCs), QDs and graphene quantum dots (GQDs) [109,119,120,121]. For instance, Wang et al. developed a sensitive fluorescence assay for the activity of casein kinase II (CK2) based on the Zr^4+^ ion-induced aggregation of phosphorylated peptide–GQDs [110]. As illustrated in Figure 12A, the peptide substrate was covalently conjugated onto the GQD surface, and the product of peptide–GQDs exhibited stable fluorescence in the presence of Zr^4+^ ions. After CK2-catalyzed phosphorylation in the presence of ATP, Zr^4+^ ions coordinated with phosphate groups on the surface of peptide–GQDs, resulting in the aggregation of GQDs. The fluorescence of GQDs was quenched through energy-transfer or electron-transfer processes. Finally, this method achieved a linear detection range from 0.1 to 1.0 unit/mL with a detection limit down to 0.03 unit/mL. However, it is complex and time-consuming to covalently couple substrate peptides onto the GQD surface. To simplify the process of preparing fluorescent probes, Song et al. developed a label-free fluorescence assay of CK2 based on peptide-capped AuNCs and Zr^4+^ ions [111]. In this work, peptide-capped AuNCs were prepared by a one-step peptide biomineralization method, and Zr^4+^ ions triggered the aggregation of phosphorylated peptide-capped AuNCs, resulting in significant quenching of the emission of AuNCs.

Upconversion nanophosphors (UCNPs) can emit intense visible emissions under the excitation of two or more near-infrared photons. The use of UCNPs as fluorescence probes can effectively suppress the auto-fluorescence of biomolecules and light scattering, improving the signal-to-background ratio. Li’s group has reported on UCNPs and Zr^4+^-coated MB-based fluorescence bioassays for the detection of hexokinase activity and PKA activity, respectively [112,122]. The excellent advantages of UCNPs make them an ideal choice as the luminescence donor in the design of luminescence resonance energy transfer (LRET) systems [113]. Liu et al. developed a sensitive and generic protein kinase assay based on a metal ion-mediated LRET [114]. As shown in Figure 12B, abundant rare earth (RE) ions on the surface of UCNPs capture phosphorylated TAMRA-labeled peptides via the interaction between RE^3+^ ions and phosphate groups. The emission of UCNPs was quenched by TAMRA through the efficient LRET process.

### 3.3. Other Metal Ion-Based Methods

As cost-effective and simple analysis methods, colorimetric methods can provide the signal of a targeted event through visual color change. In particular, noble metal nanoparticles, such as AuNPs and AgNPs, possess size-dependent LSPR absorption properties and exhibit aggregation state-dependent color changes in solution [123]. Qi et al. developed a colorimetric method for Zr^4+^ detection based on Zr^4+^-induced aggregation of ATP-stabilized AuNPs [115]. As displayed in Figure 13A, ATP was used as a capping agent to synthesize ATP-stabilized AuNPs. Zr^4+^ ions interacted with the negatively charged ATP, thus inducing the aggregation of AuNPs via the formation of a PO_3_^2−^–Zr^4+^–PO_3_^2−^ complex and causing a visual color change from red to blue. The metal ion coordination-induced aggregation of AuNPs can be regulated by PPi for indirect detection of protein kinases and phosphatases based on the difference between the affinity of PPi and phosphate (Pi). For instance, Zhang et al. developed the colorimetric assay of protein tyrosine phosphatase 1B and screening of its inhibitors based on Zr^4+^–phosphate coordination [116]. In this study, Zr^4+^ ions induced the aggregation of 4-aminophenylphosphate-modified AuNPs via the interaction between Zr^4+^ ions and phosphate groups. Protein tyrosine phosphatase 1B (PTP1B) catalyzed the hydrolysis of *p*-nitrophenyl phosphate into *p*-nitrophenol, inhibiting the aggregation of AuNPs. Deng et al. developed a real-time colorimetric assay of pyrophosphatase (PPase) via competitive coordination with Cu^2+^ between PPi and cysteine [124]. As shown in Figure 13B, Cu^2+^ ions induced the aggregation of cysteine-capped AuNPs, and the higher coordination affinity between Cu^2+^ and PPi resulted in the disaggregation of AuNPs. Based on this fact, PPase and ALP could be monitored by catalyzing the hydrolysis of PPi into Pi to inhibit the disaggregation of AuNPs. This colorimetric method had a linear detection range from 0.025 to 0.3 U with a detection limit of 0.010 U.

Nanozymes can catalyze the colorimetric reaction like natural enzymes, and their activities can be reversibly modulated by certain metal ions and PPi [125,126]. For example, Shi et al. demonstrated that PPi could recover the Cu^2+^ ion-inhibited peroxidase-like activity of AuNCs [127]. Guan et al. reported a colorimetric method for the detection of phosphates and monitoring of nucleotide phosphate-involved enzymatic hydrolysis processes [128]. As presented in Figure 14, Ce^3+^, Fe^2+^, and Cr^3+^ ions interact with AuCl^4−^/AuCl^2−^ ions on the surface of bare gold nanoparticles (BGNPs) by electrostatic interactions, greatly enhancing the peroxidase-like catalytic activity of BGNPs. Different types of phosphates generate cross-reactive signals due to the differences in the number of phosphate groups, geometry, steric effect and binding sites. Finally, the hydrolysis process of ATP was monitored by determining its metabolite.

Recently, personal glucose meters (PGMs) have been widely used in bioassays based on the enzyme-catalyzed transformation of PGM-inert sucrose (or amylon) to PGM-detectable glucose [129,130,131]. Yang et al. reported the portable and sensitive detection of PKA activity by using a commercial PGM [117]. As displayed in Figure 15, Zr^4+^-functionalized magnetic beads (ZrMBs) were employed to selectively isolate PKA-catalyzed biotin-phosphopeptide from the unphosphorylated ones. Then, SA brought biotin-invertase onto the surface of ZrMBs. The invertase catalyzed the hydrolysis of sucrose into glucose, which was monitored using a PGM. This method showed a good linear relationship in the range of 0.1 to 200 U/mL.

## 4. MOFs

Due to their chemical stability, structural flexibility, easy modification and biocompatibility, MOFs—consisting of metal nodes and organic ligands—are particularly attractive for the development of biosensors [132,133,134]. Like metal ions and Phos-tag, the dense, coordinatively unsaturated metal sites on the surface of MOFs can bind with the phosphate groups in biomolecules through multiple coordination bonds [135,136,137]. Thus, MOFs, such as Zr-based UiO-66 and PCN-222, have been widely utilized in bioassays for the detection of phosphate-containing biomolecules and related enzymes (Table 3) [138,139,140,141]. For example, Wang et al. designed a general and direct approach for immobilizing terminal phosphate-modified DNA on MOFs based on a chelation interaction and used the MOFs to achieve intracellular delivery of proteins [142,143]. Zhang et al. reported a PEC biosensor for the label-free detection of the phosphoprotein α-casein [144]. As displayed in Figure 16, PCN-222 was used as the substrate, with high porosity and tunable structures. Dopamine was added to inhibit charge recombination of electron−hole pairs and to improve the photoelectric conversion efficiency, then producing an amplified photocurrent response in the O_2_-saturated solution. In the presence of α-casein, it was directly adsorbed on PCN-222 through the coordination interaction between the phosphate groups and the inorganic Zr–O clusters. The PEC signal was significantly weakened due to the steric hindrance effect. This biosensor showed excellent sensitivity and selectivity toward α-casein with a detection limit of 0.13 μg/mL.

Nanosized MOFs can act as signal labels by loading of molecules and catalytic nanomaterials [151]. For instance, Sun et al. reported an electrochemical biosensor for the detection of glioblastoma-derived exosomes using Zr-MOFs [152]. As shown in Figure 17A, methylene blue-loaded Zr-MOFs was used to label exosomes captured by the aptamer-modified electrode through the interaction between Zr^4+^ ions and the intrinsic phosphate groups outside of exosomes. This biosensor achieved a wide detection range, from 9.5 × 10^3^ to 1.9 × 10^7^ particles/μL. Wang et al. developed a PEC immunosensor for m^6^ATP detection using [Ru(bpy)_3_]^2+^-doped UiO-66 (Ru@UiO-66) (Figure 17B) [147]. In this study, the ITO electrode was sequentially modified with black titanium dioxide (B-TiO_2_), bismuth trioxide (Bi_2_O_3_), AuNPs, MPBA and antibodies. When m^6^ATP was captured, the phosphate group of m^6^ATP was identified by Ru@UiO-66. The [Ru(bpy)_3_]^2+^ complex produced a consecutive photocurrent with the aid of AA as a sacrificial electron donor under visible light, eventually realizing the sensitive detection of m^6^ATP.

MOFs hybridized with metal nanoparticles can prevent the aggregation of metal nanoparticles and improve catalytic performance via a synergistic effect. Yan et al. reported a sensitive ECL biosensor for PKA activity analysis using AuNP- and PtNP-loaded UiO-66 (Au&Pt@UiO-66) [145]. As presented in Figure 18, Au&Pt@UiO-66 probes were immobilized onto the phosphorylated kemptide-modified electrode via the interaction between the Zr cluster on UiO-66 and the phosphate group in the phosphopeptide. Au&Pt@UiO-66 can catalyze the luminol-H_2_O_2_ ECL reaction to generate a greatly enhanced signal. Nevertheless, these methods involve the encapsulation procedures of functional molecules and nanomaterials. To simplify the preparation process, functional molecules or metal ions have been used as the components, and the prepared MOFs have been directly used as signal labels to generate strong signals for the quantification of targets. For instance, Zhang et al. prepared Zr^4+^–metalloporphyrin frameworks as three-in-one platforms for the detection of PKA activity [146].

MOFs can act as probes and/or quenchers for the design of fluorescent biosensors [153]. Among them, lanthanide MOFs (Ln-MOFs) are the most commonly used chemical fluorescence sensors for the detection of phosphate group-based molecules (e.g., ATP and phosphate) and ALP [154,155]. The competitive reaction between phosphate and metal sites can endow the MOF–DNA interaction with tunable and stimulus-responsive properties [149,156]. For example, Meng et al. reported the multiplexed detection and imaging of intracellular mRNA based on the phosphate-stimulated release of dye-labeled DNA from MOFs [157]. Yu et al. developed a molecular sensing and logic system based on phosphate-modulated MOF–DNA interaction [148]. As shown in Figure 19, Zr-based PCN-224 adsorbs dye-labeled ssDNA via the formation of Zr–O–P bonds and quenches the fluorescence through the FRET process. Inorganic monophosphate (Pi), pyrophosphate (PPi), tripolyphosphate (STPP) and hexametaphosphate (SHMP) coordinate to Zr^4+^ ions in the clusters of PCN-224 and prevent the subsequent interaction between PCN-224 and DNA, thus blocking the FRET process. Based on the different site-occupying abilities of STPP and protonated Pi, inorganic pyrophosphatase (PPase) and ALP were sensitively determined.

MOFs with enzyme-like features have received considerable attention in biosensing. Phosphate-containing species can interact with metal ions on the surface of MOFs to modulate their catalytic activities [158,159,160,161]. A series of MOFs such as MIL-53(Fe), Cu-MOFs and Fe-MIL-88A have been applied for colorimetric detection of ALP, Pi and thrombin based on the interaction between phosphate and MOFs [162,163,164]. Wang et al. constructed a Cu-MOF-based colorimetric logic gate for ALP detection [165]. As displayed in Figure 20A, PPi binds to Cu^2+^ ions and inhibits the Cu-MOF-catalyzed oxidation of TMB by H_2_O_2_. In the presence of ALP, PPi is hydrolyzed into Pi, losing its inhibition ability due to the low affinity of Pi and Cu^2+^. In most of the previously reported MOF nanozymes, metal nodes acted as the active centers. Wang et al. found that the ligand of 2,2′-bipyridyl-4,4′-dicarboxylicacid could endow Zr-based MOFs with a peroxidase-mimicking ability. Based on this fact, they developed a colorimetric sensing array to quantify the phosphoprotein α-casein (Figure 20B) [150]. In this work, Zr nodes in MOFs acted as the specific sites to recognize phosphate groups in proteins. Phosphoprotein adsorbed on the surface of MOFs inhibited the peroxidase-mimicking activity and suppressed the chromogenic reaction.

## 5. Conclusions

In this review, we summarized progress in the development of biosensors based on metal ion–phosphate chelation interactions for molecular recognition. A broad array of Phos-tag-based derivatives and nanomaterials have been used as antibody alternatives in specific labeling of phosphorylation sites. Metal ions such as Zr^4+^ ions can act as the bridge to link different signal-reporting species to the phosphorylation sites. Moreover, MOFs with abundant coordinatively unsaturated metal sites have been adapted to various biosensors for sensitive and facile signal amplification with flexible assay designs. Despite impressive achievements in the detection of phosphate-containing biomolecules and related enzymes, there are still several shortcomings to be addressed. For example, most biosensors are used to detect the targets in serum samples or cell lysates, and only a few of them can work in living cells. The design of new nanomaterials with outstanding signal amplification ability may remarkably improve the performance of biosensors. Although MOFs with dense coordinatively unsaturated metal sites on their surface can bind with phosphate groups through multiple coordination bonds, they always show low structural stability and barely bind to the substrates in a rapid and controllable manner. Moreover, the performance of MOF-based biosensors can be significantly affected by the morphology, size, and structure of MOFs. For practical applications such as point-of-care diagnosis, biosensors should have long-term stability, potential to be miniaturized and simple operating procedures. Although the electrochemical Phos-tag-based biosensors have achieved great advancements in disease diagnosis and treatment, they still suffer from several challenges of complex surface modification, sample preparation and low reproducibility. With the innovations of sensitive detection instruments and advanced nanotechnology, we believe that biosensors based on metal ion–phosphate chelation interactions for molecular recognition will be successfully applied for health monitoring in the future.

## Figures and Tables

**Figure 1 molecules-28-04394-f001:**
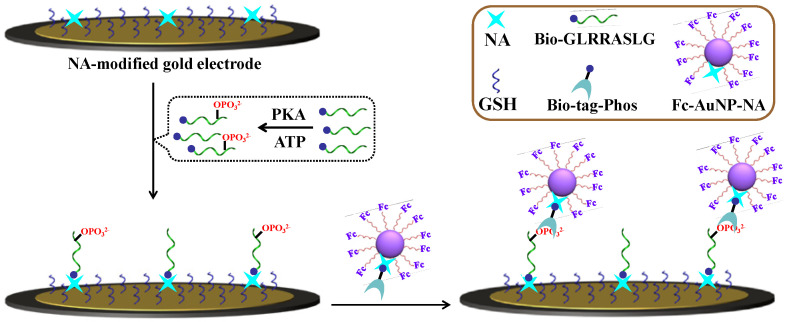
Schematic illustration of the electrochemical method for PKA detection by the signal amplification of Fc-AuNP-NA-Bio-tag-Phos conjugate [43]. Copyright 2021 Elsevier.

**Figure 2 molecules-28-04394-f002:**
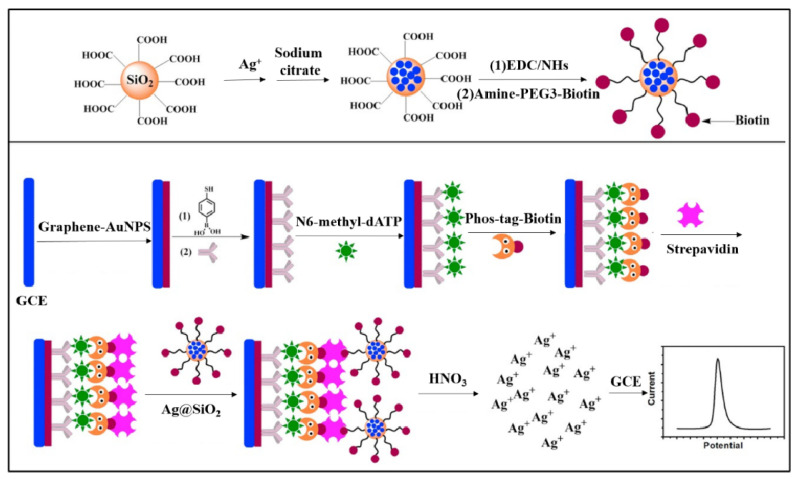
Schematic illustration of preparation of Ag@SiO_2_ and biosensor fabrication and electrochemical detection of m6A [44]. Copyright 2017 Elsevier.

**Figure 3 molecules-28-04394-f003:**
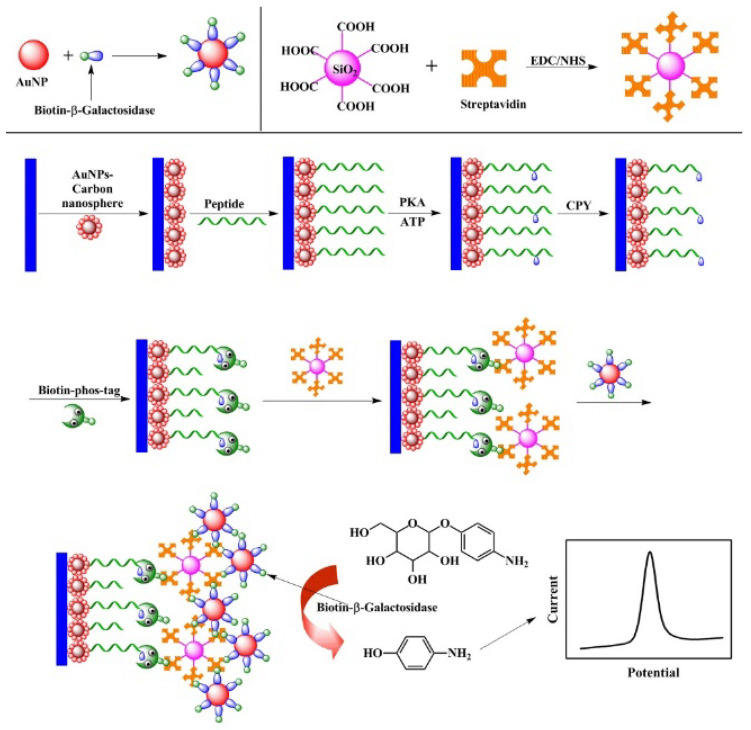
Schematic illustration of the biosensor fabrication process and *β*-galactosidase-based PKA detection strategy [47]. Copyright 2021 Elsevier.

**Figure 4 molecules-28-04394-f004:**
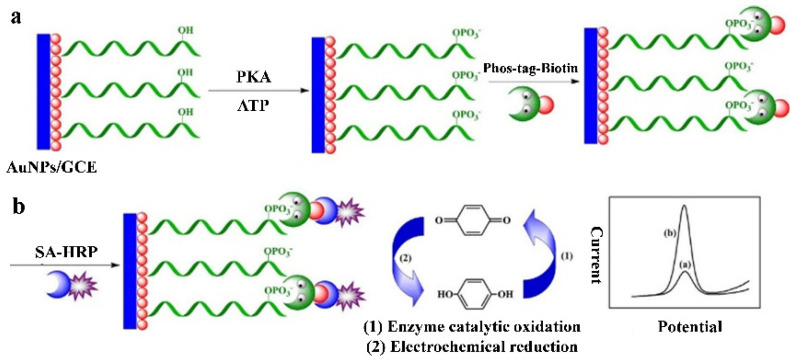
Schematic illustration of the Phos-tag-biotin–streptavidin-HRP system-based electrochemical biosensor for PKA activity assay [48]. Copyright 2015 Elsevier.

**Figure 5 molecules-28-04394-f005:**
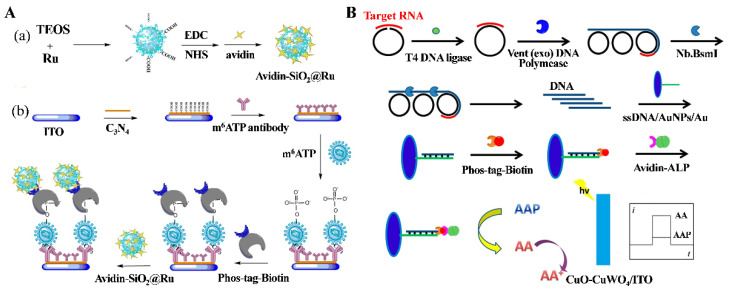
(**A**) Schematic illustration of (**a**) preparation of avidin-Ru@SiO_2_ and (**b**) PEC biosensor fabrication [53]. Copyright 2018 Elsevier. (**B**) Schematic illustration of ALP-based PEC biosensor for microRNA detection [56]. Copyright 2018 Elsevier.

**Figure 6 molecules-28-04394-f006:**
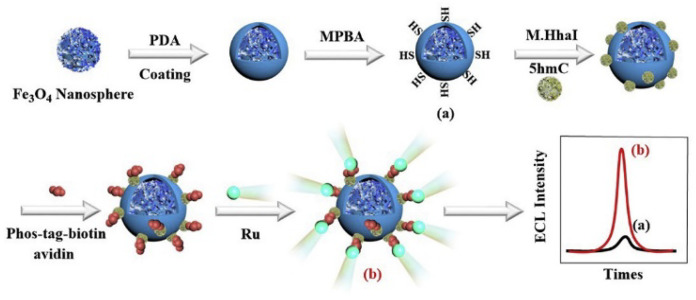
Schematic illustration of the ECL biosensor fabrication process and 5 hmC detection using Phos-tag-biotin and Ru complex [57]. Copyright 2020 Elsevier.

**Figure 8 molecules-28-04394-f008:**
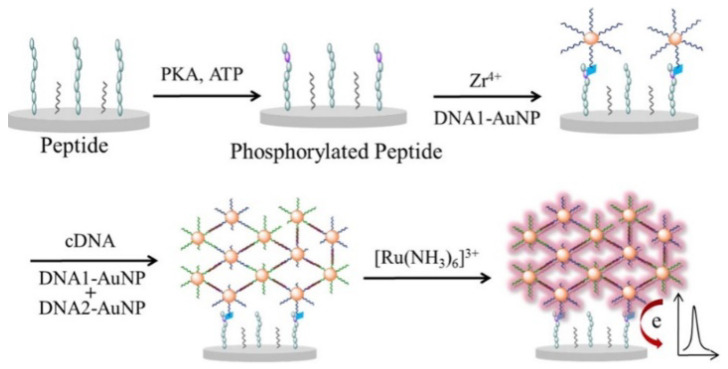
Schematic illustration of DNA-AuNP-assembled polymeric network-amplified electrochemical biosensor for kinase activity detection [97]. Copyright 2014 American Chemical Society.

**Figure 9 molecules-28-04394-f009:**
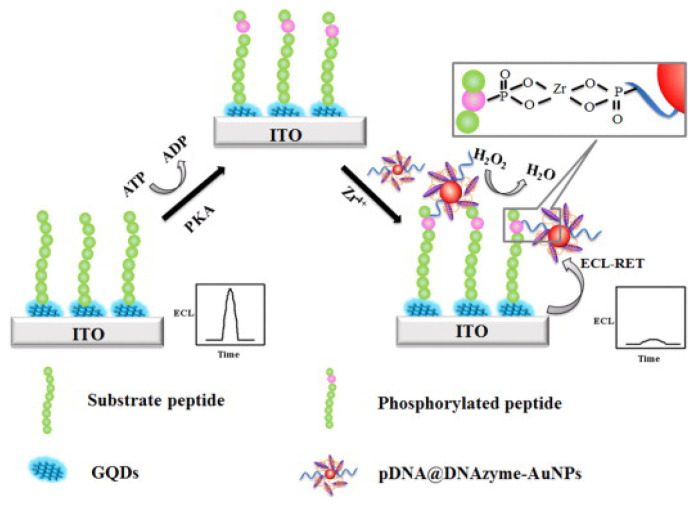
Schematic illustration of ECL assay for PKA based on double-quenching of GQDs by DNA@DNAzyme-AuNPs [100]. Copyright 2016 American Chemical Society.

**Figure 10 molecules-28-04394-f010:**
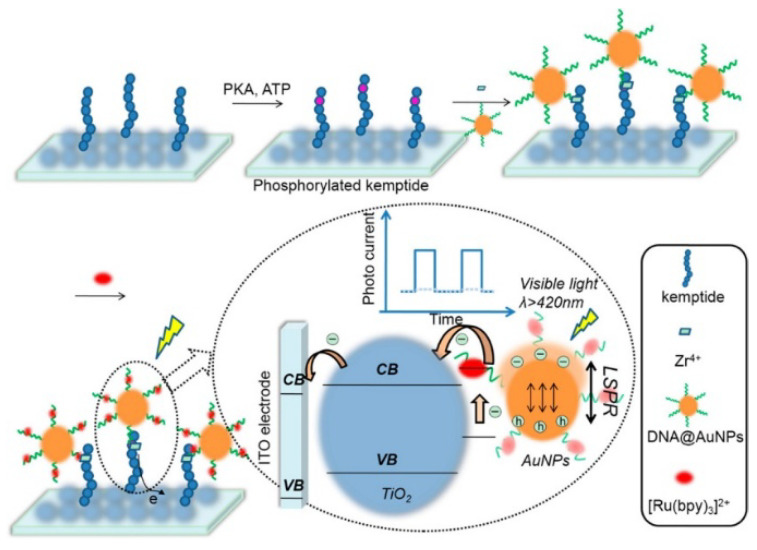
Schematic illustration of DNA-AuNP-based PEC biosensor for kinase activity detection with the aid of Zr^4+^ ions [103]. Copyright 2016 American Chemical Society.

**Figure 11 molecules-28-04394-f011:**
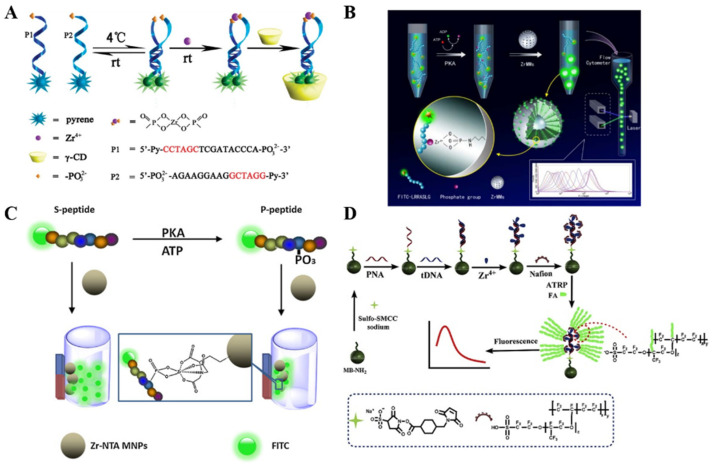
(**A**) Schematic illustration of molecular beacon-based sensing system for sensitive detection of Zr^4+^ using γ-CD as a signal amplifier [104]. Copyright 2012 American Chemical Society. (**B**) Schematic illustration of the ZrMM-based cytometric bead assay for PKA detection [106]. Copyright 2013 American Chemical Society. (**C**) Schematic illustration of the fluorescence kinase activity assay based on Zr-NTA-MNP enrichment [107]. Copyright 2013 Elsevier. (**D**) Schematic illustration of the biosensor for lung cancer DNA detection based on phosphate-Zr^4+^-sulfonate chemistry and atom transfer radical polymerization [108]. Copyright 2020 Elsevier.

**Figure 12 molecules-28-04394-f012:**
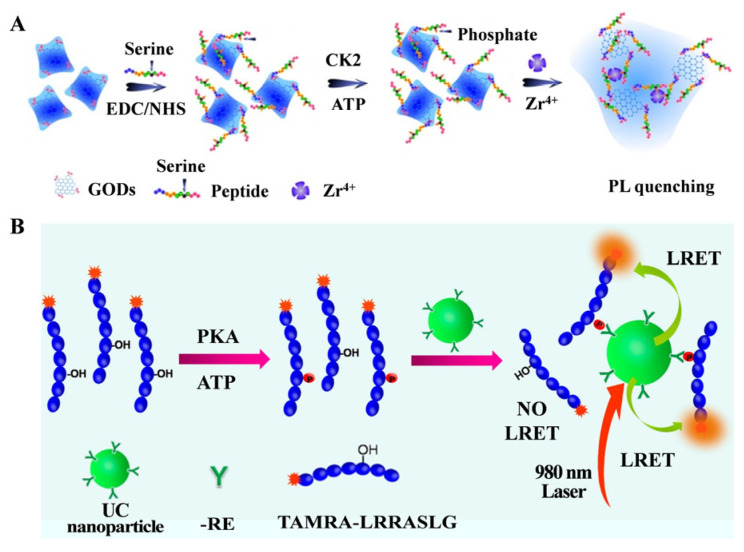
(**A**) Schematic illustration of the CK2 kinase assay based on aggregation and fluorescence quenching of phosphorylated peptide–GQDs via Zr^4+^ ion linkage [110]. Copyright 2013 American Chemical Society. (**B**) Schematic illustration of design principle of the UCNP-based LRET assay for the detection of PKA activity [114]. Copyright 2014 American Chemical Society.

**Figure 13 molecules-28-04394-f013:**
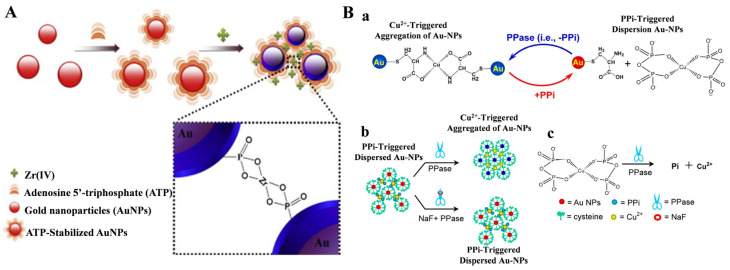
(**A**) Schematic illustration of colorimetric detection of Zr^4+^ ions based onZr^4+^-induced aggregation of ATP-stabilized AuNPs [115]. Copyright 2013 Elsevier. (**B**) Schematic illustration of the principle for (**a**) reversible competitive coordination chemistry regulated by PPase, (**b**) colorimetric detection of PPase activity and the inhibition efficiency of NaF, and (**c**) the catalytic reaction of PPase [124]. Copyright 2013 American Chemical Society.

**Figure 14 molecules-28-04394-f014:**
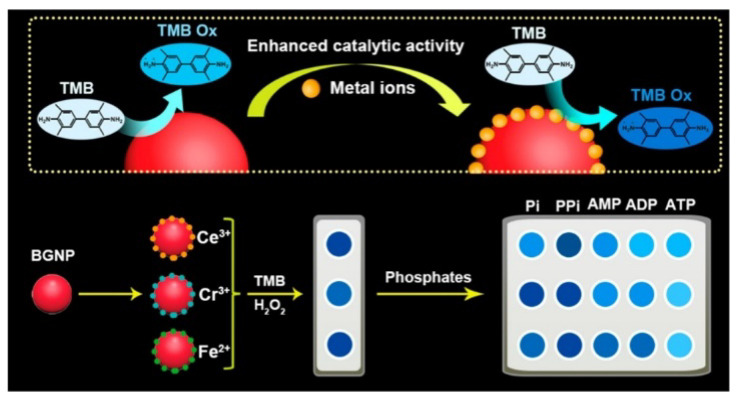
Schematic illustration of mechanism of the colorimetric sensor array for the detection of phosphates and monitoring of nucleotide phosphate-involved enzymatic hydrolysis processes based on the peroxidase-like activity of BGNP−metal ion ensembles [128]. Copyright 2021 American Chemical Society.

**Figure 15 molecules-28-04394-f015:**
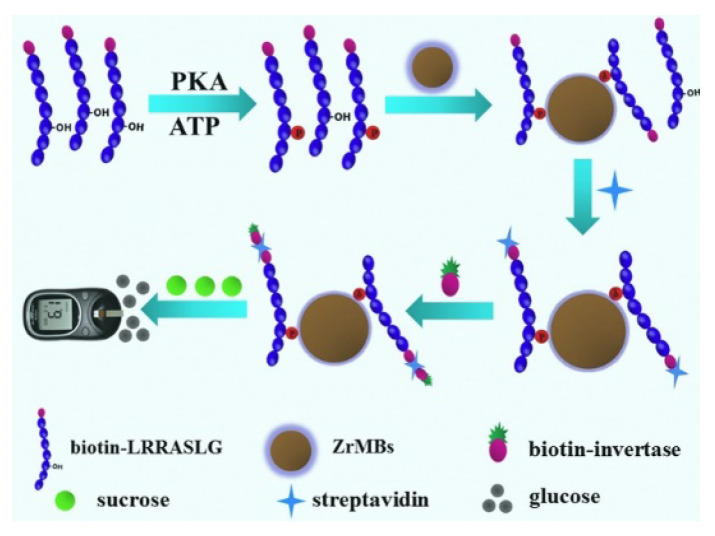
Schematic illustration of detection principle of the PGM-based portable PKA assay [117]. Copyright 2015 Elsevier.

**Figure 16 molecules-28-04394-f016:**
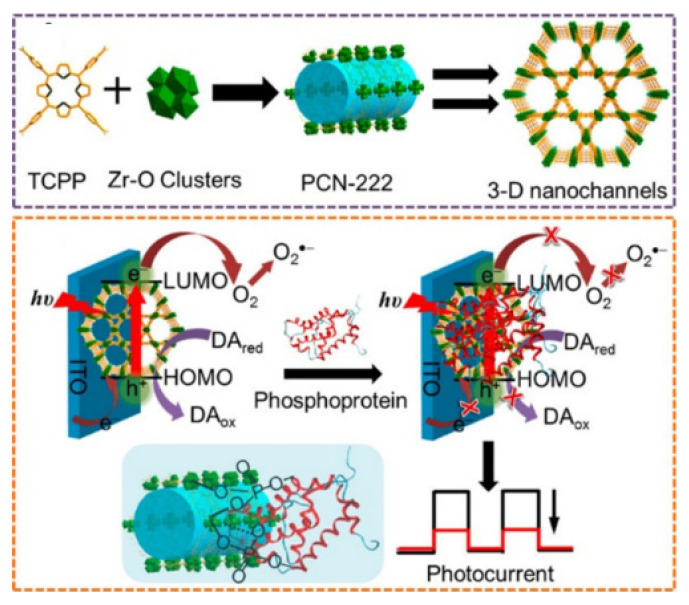
Schematic illustration of the construction of PCN-222 and the mechanism of charge recombination suppression-based PEC strategy for detection of phosphoprotein [144]. Copyright 2016 American Chemical Society.

**Figure 17 molecules-28-04394-f017:**
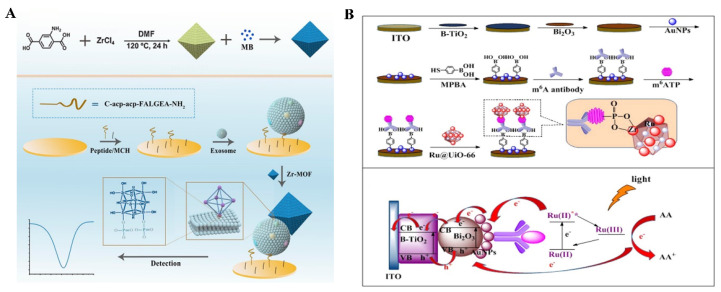
(**A**) Schematic illustration of the fabrication process of methylene blue-loaded Zr-MOF-based nanoprobes and the principle of the electrochemical biosensor for the detection of exosomes [152]. Copyright 2020 American Chemical Society. (**B**) Schematic illustration of the fabrication procedure and mechanism of photocurrent generation of Ru complex-doped Zr-MOF-based ECL immunosensor for the detection of N^6^-methyladenine [147]. Copyright 2019 Elsevier.

**Figure 18 molecules-28-04394-f018:**
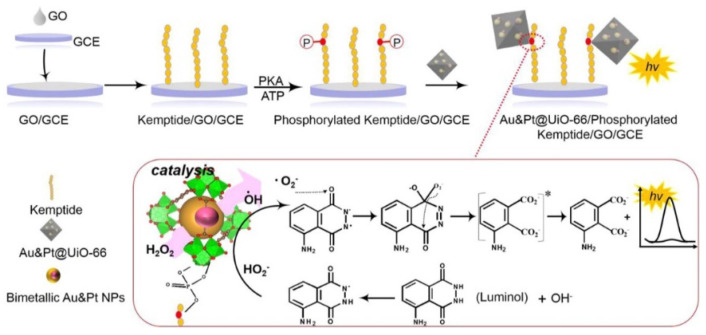
Schematic illustration of the configuration of ECL biosensor for kinase activity detection and the mechanism of enhanced luminol ECL catalyzed by Au&Pt@UiO-66 [145]. Copyright 2019 Elsevier.

**Figure 19 molecules-28-04394-f019:**
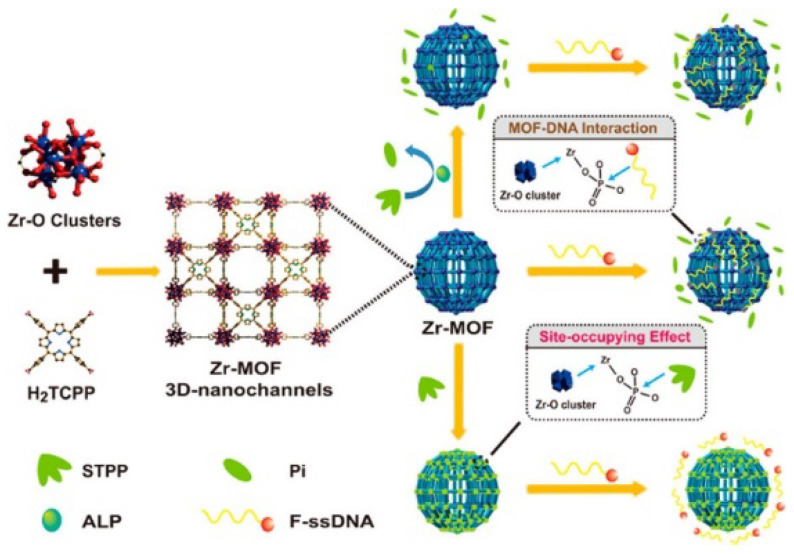
Schematic illustration of site-occupying effect-modulated MOF–DNA interaction via Zr–O–P bonds [148]. Copyright 2020 American Chemical Society.

**Figure 20 molecules-28-04394-f020:**
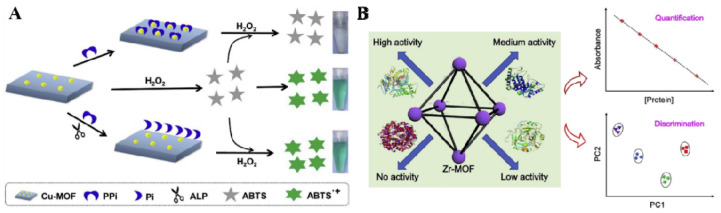
(**A**) Schematic illustration of colorimetric sensing strategy for ALP activity based on PPi-modulated catalytic activity of Cu-MOFs [165]. Copyright 2018 Elsevier. (**B**) Schematic illustration of the peroxidase-like Zr-MOF-based colorimetric sensor array for the quantification and discrimination of phosphorylated proteins [150]. Copyright 2020 Elsevier.

**Table 1 molecules-28-04394-t001:** Comparison of the detection performances of Phos-tag-based biosensors.

Method	Substrate	Signal Report	Target	Linear Range	Detection Limit	Ref.
EC	Avidin	Fc/neutravidin-AuNPs	PKA	5 × 10^−3^–20 U/mL	5 × 10^−3^ U/mL	[43]
EC	Antibody	Avidin/AgNPs-SiO_2_	m6A	0.2–5 × 10^2^ nM	7.8 × 10^−2^ nM	[44]
EC	DNA/AuNPs	Biotin-ALP	PNK	0–5 U/mL	2.75 × 10^−3^ U/mL	[45]
EC	Antibody-graphene	Avidin-ALP	5hmC	0.1–30 nM	3.25 × 10^−2^ nM	[46]
EC	Peptide/AuNPs/CNs	SA-SiO_2_/*β*-galactosidase	PKA	5 × 10^−2^–1 × 10^2^ U/mL	1.4 × 10^−2^ U/mL	[47]
EC	AuNPs	Avidin-HRP	PKA	0.5–25 U/mL	0.15 U/mL	[48]
PEC	Antibody-g-C_3_N_4_/CdS	Avidin-CuO	RNA	1 × 10^−2^–10 nM	3.53 pM	[49]
PEC	Antibody-β-CD-GO/Fe_3_O_4_	PAMAM-avidin-ALP	5hmC	1 × 10^−2^–50 nM	3.2 pM	[50]
PEC	Peptide-g-C_3_N_4_/AuNPs	Avidin-ALP	PKA	5 × 10^−2^–1 × 10^2^ U/mL	1.5 × 10^−2^ U/mL	[51]
PEC	Peptide-Bi_2_S_3_	SA	PKA	5 × 10^−2^–1 × 10^2^ U/mL	1.7 × 10^−2^ U/mL	[52]
PEC	Antibody-g-C_3_N_4_/CdS	Avidin/Ru-SiO_2_	m6A	1 × 10^−2^–10 nM	3.23 pM	[53]
PEC	AuNPs/TiO_2−x_/MoS_2_	Avidin-ALP	m6A	0.3–1 × 10^2^ nM	0.14 nM	[54]
PEC	AuNPs/MoS_2_/graphene	SA-*β*-galactosidase	HAT	0.3–1 × 10^2^ nM	0.14 nM	[55]
PEC	CuO-CuWO_4_	Avidin-ALP	miRNA	1 × 10^−6^–0.1 nM	0.47 fM	[56]
ECL	MPBA-PDA@Fe_3_O_4_	Ru-avidin	5hmC	1 × 10^−2^–5 × 10^2^ nM	2.86 pM	[57]
ECL	Antibody-Fe_3_O_4_@SiO_2_/GO	Avidin/Ru-PAMAM	5hmC	0.1–30 nM	4.7 × 10^−2^ nM	[58]
FL	-	FITC/Cy5-peptide	PKA	1 × 10^−4^–10 nM	1 fM	[59]
FL	SA-coated MBs	Cy5/BHQ2-DNA	PKA	1 × 10^−4^–10 U/mL	1.98 × 10^−5^ U/mL	[60]

Abbreviations: EC, electrochemistry; Fc, ferrocene; AuNPs, gold nanoparticles; PKA, protein kinase A; m6A, N6-methyladenosine; AgNPs, silver nanoparticles; SiO_2_ NPs, silica nanoparticles; ALP, alkaline phosphatase; CNs, carbon nanosphere; PNK, T4 polynucleotide kinase; 5hmC, 5-hydroxymethylcytosine; HRP, horseradish peroxidase; PDA, polydopamine; MPBA, mercaptobenzoic acid; FL, fluorescence; PAMAM, polyamidoamine dendrimers; HAT, histone acetyltransferase; g-C_3_N_4_, graphite-like carbon nitride; GO, graphene oxide; SA, streptavidin; FITC, fluorescein.

**Table 3 molecules-28-04394-t003:** Comparison of the detection performance of biosensors based on MOF-phosphate interactions.

Method	Substrate	Signal Report	Target	Linear Range	Detection Limit	Ref.
ECL	DNA	Eu@Zr-MOF	Mucin 1	1.13 × 10^−4^–0.113 ng/mL	0.37 fg/mL	[139]
ECL	Zr-PCN-222	–	Protein	0–2 μg/mL	0.13 μg/mL	[144]
ECL	Kemptide-GO	AuNPs-PtNPs@Zr-UiO-66	PKA	1.5 × 10^−2^–0.25 U/mL	9 mU/mL	[145]
ECL	Kemptide-AuNPs	Zr-ZnTCPP-MOF-525	PKA	1 × 10^−2^–20 U/mL	5 mU/mL	[146]
PEC	AuNPs/Bi_2_O_3_/B-TiO_2_	Ru@Zr-UiO-66	m6A	5 × 10^−2^–30 nM	16.7 pM	[147]
FL	Zr-PCN-224	TAMRA-DNA	ALP	0.5−10 and 10−100 U/L	0.19 U/L	[148]
FL	Ce-MOFs	FITC-DNA	ALP	2−1 × 10^2^ U/L	0.18 U/L	[149]
Color	–	Zr-MOFs	α-casein	0.17–5 μg/mL	0.16 μg/mL	[150]

Abbreviation: AuNPs, gold nanoparticles; ZnTCPP, zinc tetrakis(carboxyphenyl)-porphyrin; B-TiO_2_, black titanium dioxide; PKA, protein kinaseA; ALP, alkaline phosphatase.

## Data Availability

Not applicable.

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
