# Peer review of "Biosensors with Metal Ion–Phosphate Chelation Interaction for Molecular Recognition"

_molecules, 2023, doi:10.3390/molecules28114394_

Round 1

Reviewer 1 Report

 Paper entitled Biosensors with metal ion-phosphate chelation interaction for molecular recognition by Ma et al. summarizes recent trends in molecular recognition based on phosphates and their ability to form complexes with metal ions. Topic is interesting and the paper can be published in Molecules after a major revision.

Paragraph 2. Phos-tag does not address ion-phosphate chelation interactions. It is not clear, why it is a part of this review.

Lines 83-87

Heterogeneous assays exhibit high sensitivity, but the steric hindrance effect in such methods may limit the interaction between enzyme and substrate immobilized on the solid surface. For this consideration, we reported an electrochemical method for the detection of protein kinase A (PKA) by converting homogeneous analysis into heterogeneous assay through avidin-biotin interaction.

I do not understand this conclusion. Authors said, that heterogeneous assays are affected by steric hindrance so they used heterogeneous assay to solve this problem.

Font in all figures is too small. It is not possible to read the description inserted in the picture. What is ATP, GSH, NA, etc., in Figure 1. Figures should be self-standing. It refers to all figures in the manuscript.

Line 132: PEC methods – abbreviation should be explained

Why is PKA activity determination important? Why detection of polynucleotide kinase (PNK) and 5-hydroxymethylcytosine I is important? This question refers to all examples given in the manuscript. It is very important to show in the review that the topic is important. Good reviews are not just a short summary of the manuscripts on the topic. It should be justified, that it is important.

 Line 236: It has been reported that transition metal ions such as Fe3+, Ga3+ and Dy3+

To be more specific, Ga and Dy are lanthanides.

 4. Metal-organic frameworks: it should be shortly explained what are MOFs.

Extensive editing of English language is required

Author Response

We thank the reviewer for his/her positive comments: "Paper entitled Biosensors with metal ion-phosphate chelation interaction for molecular recognition by Ma et al. summarizes recent trends in molecular recognition based on phosphates and their ability to form complexes with metal ions. Topic is interesting and the paper can be published in Molecules after a major revision."

Comment 1: "Paragraph 2. Phos-tag does not address ion-phosphate chelation interactions. It is not clear, why it is a part of this review."

Response: We have added the following sentences in the Part 2: "The dinuclear Zn(II) complex of 1,3-bis[bis(pyridin-2-ylmethyl)-amino]propan-2-olato named Phos-tag has been considered as a prevalent phosphate-binding tag. The complex with a vacancy on the two Zn(II) ions can specifically chelate with phosphate monoester dianion in the phosphorylated peptide or protein. The formed 1:1 complex shows a dissociation constant of approximately 10-8 M at neutral pH value [31]. The affinity for phosphate monoester dianion is more than 10,000 times stronger than that for other anions. The value is close to that of the interaction between anti-phosphorylation antibody and phosphorylated amino acid residues. Because of its small size, Phos-tag shows negligible effect on the structure around the phosphorylation site."

Comment 2: "Lines 83-87 Heterogeneous assays exhibit high sensitivity, but the steric hindrance effect in such methods may limit the interaction between enzyme and substrate immobilized on the solid surface. For this consideration, we reported an electrochemical method for the detection of protein kinase A (PKA) by converting homogeneous analysis into heterogeneous assay through avidin-biotin interaction. I do not understand this conclusion. Authors said, that heterogeneous assays are affected by steric hindrance so they used heterogeneous assay to solve this problem."

Response: In this work, the phosphorylation reaction occurred in a homogeneous solution and the detection assay was carried out at a solid electrode surface. The biosensor was designed by integrating the merits of both homogeneous reaction and heterogeneous detection. We have revised the presentations.

Comment 3: "Font in all figures is too small. It is not possible to read the description inserted in the picture. What is ATP, GSH, NA, etc., in Figure 1. Figures should be self-standing. It refers to all figures in the manuscript."

Response: We have improved the quantification of the re-used figures.

Comment 4: "Line 132: PEC methods – abbreviation should be explained"

Response: The full name of PEC has been shown in Introduction.

Comment 5: "Line 236: It has been reported that transition metal ions such as Fe3+, Ga3+ and Dy3+. To be more specific, Ga and Dy are lanthanides."

Response: Yes, we have revised the presentation.

Comment 6: "Metal-organic frameworks: it should be shortly explained what are MOFs."

Response: We have defined it in Introduction and Part 4.

Comment 7: "Extensive editing of English language is required."

Response: We have the manuscript checked by a native English-speaking colleague.

Reviewer 2 Report

The author summarizes the development of biosensors with metal ion-phosphate chelation interaction for molecular recognition. This review is classified into three parts according to the type of recognition elements toward phosphate-containing targets: Phos-tag, metal ions and metal-organic frameworks (MOFs). The author's summary is comprehensive and detailed, and the terminology is fully explained to make it easy to understand. However, the manuscript still needs to be revised before publication. My specific comments are as follows:

1.       Please check the picture format, some pictures are not in the middle.

2.       Please note that the valid numbers and units are uniform, especially the 0.0001to 0.2 in line 462 and  in tables.

3.       The author should add academic opinions and judgments about the future development prospect of biosensors with metal ion-phosphate chelation interaction for molecular recognition.

4.       Some disadvantages of Phos-tag, metal ions and metal-organic frameworks can be summarized objectively in line 587.

5.       The text in the Figure 15C is too small to see.

6.       It could be better to summarize the contrasting characteristics of all types of biosensors.

 Please check again for each reference carefully because there were some format errors in the references. Especially the format of the spaces.

Author Response

We thank the reviewer for his/her positive comments: "The author summarizes the development of biosensors with metal ion-phosphate chelation interaction for molecular recognition. This review is classified into three parts according to the type of recognition elements toward phosphate-containing targets: Phos-tag, metal ions and metal-organic frameworks (MOFs). The author's summary is comprehensive and detailed, and the terminology is fully explained to make it easy to understand. However, the manuscript still needs to be revised before publication. My specific comments are as follows:"

Comment 1: "Please check the picture format, some pictures are not in the middle."

Response: The picture format was designed according to the journal style. We have checked all the figures carefully.

Comment 2: "Please note that the valid numbers and units are uniform, especially the 0.0001 to 0.2 in line 462 and  in tables."

Response: We have revised the formats of valid numbers and units.

Comment 3: "The author should add academic opinions and judgments about the future development prospect of biosensors with metal ion-phosphate chelation interaction for molecular recognition."

Response: We have rewritten the conclusions with a brief discussion of the pros and cons of the approach.

Comment 4: "Some disadvantages of Phos-tag, metal ions and metal-organic frameworks can be summarized objectively in line 587."

Response: We have added the comments about the disadvantages of biosensors based on metal ion-phosphate chelation interaction.

Comment 5: "The text in the Figure 15C is too small to see."

Response: We have improved the quantification of the re-used figures.

Comment 6: "It could be better to summarize the contrasting characteristics of all types of biosensors."

Response: The characteristics of all types of biosensors have been addressed in Part 2.

Comment 7: "Please check again for each reference carefully because there were some format errors in the references. Especially the format of the spaces."

Response: We have checked and revised the reference format carefully.

Reviewer 3 Report

The authors review the progress in the design of biosensors based on metal ion-phosphate chelation interaction. The work is interesting and no review paper on this topic was found in Web of Science. I think the work is suitable for publication in Molecules after modification.

Comments:

1. The sensing techniques in this view mainly include electrochemistry, fluorescence and colorimetry. The principle and application fields of each technique should be briefly discussed at the begin of each part.

2. The binding affinity for metal ion-phosphate interaction should be added if it was reported.

3. A brief discussion of the advantages and disadvantages of the biosensors should be added.

4. In this work, the authors summarized the progress in the development of biosensors based on metal ion-phosphate chelation interaction for molecular recognition. Thus, to give readers a comprehensive understand of metal ion-phosphate chelation interaction, some recent progress including  Colorimetric sensor based on V-phosphate chelation interaction (Small 2023, 19, 2206465); Electrochemical biosensor based on Zr-phosphate chelation interaction (Biosens. Bioelectron. 2021, 112907) deserve citation.

Author Response

We thank the reviewer for his/her positive comments: "The authors review the progress in the design of biosensors based on metal ion-phosphate chelation interaction. The work is interesting and no review paper on this topic was found in Web of Science. I think the work is suitable for publication in Molecules after modification."

Comment 1: The sensing techniques in this view mainly include electrochemistry, fluorescence and colorimetry. The principle and application fields of each technique should be briefly discussed at the begin of each part.

Response: The characteristics and applications of all types of biosensors have been addressed in Part 2.

Comment 2: The binding affinity for metal ion-phosphate interaction should be added if it was reported.

Response: We have added the reported binding affinity in the revised manuscript.

Comment 3: A brief discussion of the advantages and disadvantages of the biosensors should be added.

Response: We have rewritten the conclusions with a brief discussion of the pros and cons of the approach.

Comment 4: In this work, the authors summarized the progress in the development of biosensors based on metal ion-phosphate chelation interaction for molecular recognition. Thus, to give readers a comprehensive understand of metal ion-phosphate chelation interaction, some recent progress including  Colorimetric sensor based on V-phosphate chelation interaction (Small 2023, 19, 2206465); Electrochemical biosensor based on Zr-phosphate chelation interaction (Biosens. Bioelectron. 2021, 112907) deserve citation.

Response: We have updated the references.

Round 2

Reviewer 1 Report

I accept your answers.

English has been improved.